Attribute based honey encryption algorithm for securing big data: Hadoop distributed file system perspective

Kapil Gayatri gayatri1258@gmail.com 1
Agrawal Alka 1
Attaallah Abdulaziz 2
Algarni Abdullah 2
Kumar Rajeev rs0414@gmail.com 1
Khan Raees Ahmad 1
1 Information Technology, Babasaheb Bhimrao Ambedkar University , Lucknow , Uttar Pradesh , India
2 Faculty of Computing and Information Technology, King Abdulaziz University , Jeddah , Saudi Arabia
Dolev Shlomi
Electronic publication date: 2020 Feb 17
Publication date: 2020
Volume: 6
Electronic Location ID: e259
Received 2019 May 22; Accepted 2020 Jan 21
Copyright: ©2020 Kapil et al.
Copyright year: 2020
Copyright holder: Kapil et al.
License: This is an open access article distributed under the terms of the Creative Commons Attribution License, which permits unrestricted use, distribution, reproduction and adaptation in any medium and for any purpose provided that it is properly attributed. For attribution, the original author(s), title, publication source (PeerJ Computer Science) and either DOI or URL of the article must be cited.
License URL: https://creativecommons.org/licenses/by/4.0/

Keywords: Big data, Data security, And encryption-decryption, HDFS, Hadoop, Cloud storage

Funding: Council of Science & Technology, Uttar Pradesh, India This work is sponsored by Council of Science & Technology, Uttar Pradesh, India under F. No. CST/D-2408. The funders had no role in study design, data collection and analysis, decision to publish, or preparation of the manuscript.

==============================
Hadoop has become a promising platform to reliably process and store big data. It provides flexible and low cost services to huge data through Hadoop Distributed File System (HDFS) storage. Unfortunately, absence of any inherent security mechanism in Hadoop increases the possibility of malicious attacks on the data processed or stored through Hadoop. In this scenario, securing the data stored in HDFS becomes a challenging task. Hence, researchers and practitioners have intensified their efforts in working on mechanisms that would protect user’s information collated in HDFS. This has led to the development of numerous encryption-decryption algorithms but their performance decreases as the file size increases. In the present study, the authors have enlisted a methodology to solve the issue of data security in Hadoop storage. The authors have integrated Attribute Based Encryption with the honey encryption on Hadoop, i.e., Attribute Based Honey Encryption (ABHE). This approach works on files that are encoded inside the HDFS and decoded inside the Mapper. In addition, the authors have evaluated the proposed ABHE algorithm by performing encryption-decryption on different sizes of files and have compared the same with existing ones including AES and AES with OTP algorithms. The ABHE algorithm shows considerable improvement in performance during the encryption-decryption of files.

Introduction

Data security has now become one of the top most concerns for any individual or organization. Day by day, substantial amount of information is transferred through digital applications which require heaps of extra storage space, processing assets and dynamic framework execution. The exponential use of smart phones, social networking sites, downloaded apps, web sensor are generating huge amount of data. This has led to several issues in big data including storage customization, security, cost-effectiveness, smooth performance, vendor lock-in, and compliance. All these issues have their importance in Hadoop. However, big data security and privacy has become the burning issue for Hadoop HDFS data storage and distributed computing. This study essentially focuses on ensuring security and privacy for big data at the storage level.

When utilizing the Hadoop HDFS data storage service, clients have no compelling reason to store information locally and thus convey it constantly. As is the usual norm, the information is kept on the Hadoop HDFS storage server to ensure that clients can access a given information as per their convenience, irrespective of the time and place they choose to avail it from. The Hadoop HDFS storage server provides both hardware allocation and information security assurance. As long as the clients are connected with the internet, they get their information easily. Hadoop is an on-going innovation which is utilized as a system for the huge information storage. It is an open source execution of the structure dependent on java. Hadoop is utilized in a substantial bunch or as an open cloud administration. This process is termed as the standard conveyance parallel processing framework (Polato et al., 2014). The versatility of Hadoop is evident by its ubiquitious use, yet Hadoop is devoid of effective mechanisms to ward off security breaches of the data stored in HDFS

As Hadoop provides no inherent security for the information stored in it, numerous methods and approaches for securing the stored HDFS files have been explored by various researchers and practitioners. Among all these efforts, encryption seems to be the most promising answer for securing information in HDFS that is kept in DataNodes as well as for securely exchanging datafrom one DataNode to another DataNode while executing MapReduce jobs. Encryption techniques can considerably reduce the security breaches and data infringement in Hadoop environment. However, the results obtained through various encryption algorithms have demonstrated that the document sizes of the original files can be extended to about one and a half. Further, the uploading as well as the downloading time of a given file can also be increased. Hence, to adress these concerns, the researchers of this study have propositioned a new encryption-decryption algorithm, i.e., the ABHE. As per the simulation results, this technique has shown marked improvements over encryption-decryption time in comparison with the already available algorithms including the Advanced Encryption Standard (AES) and AES with OTP (Mahmoud, Hegazy & Khafagy, 2018).

The main contributions of paper are:

• To carry out the in-depth study of big data processor, i.e., Hadoop and to assess its strength and weakness in terms of security and privacy;

• To propose an ABHE, a secure and efficient algorithm executed on single and two DataNodes in Hadoop. Also, it ensures the full security against all side channel attacks, brute force attack and collusion attack;

• To conduct experiments on test data to prove the efficacy of our proposed approach ABHE vs. other secure approches i.e., AES and AES-OTP;

• The performance of proposed ABHE has been calculated in terms of File size, Encryption Time, Decryption Time, Throughput and Power Consumption;

• The result shows that ABHE improves the execution time (total time taken for encryption and decryption of data) without affecting the size of original file.

The rest of the paper has been divided into the following sections:

• Section 2- enlists the pertinent research done in the domain of Big Data storage;

• Section 3- enunciates the suggested data encryption algorithm formulated on ABHE;

• Section 4- presents the integration of the suggested algorithm with Hadoop environment. Furthermore, this section also provides a comparison between the efficacy of the suggested encryption approach vis-a-vis the two already available encryption algorithms namely; AES and AES with OTP with different sizes of text files ranging from MBs to GBs (64 MB, 128 MB, 512 MB, and 1 GB);

• Section 5- underlines the significance of this research study;

• Section 6- concludes the study.

Related work

Hadoop security

Hadoop was created in 2008 with the intention to manage only huge amount of data confined to a specific condition. Thus, security issues weren’t the topmost preference (Yalla et al., 2016). For any data storage, Hadoop employs the user’s name. In the default node, there is no encryption among Hadoop, the client host as well as the HDFS. All the records are feeded into and constrained by a central server which is known as NameNode. Thus, HDFS lacks in security system against capacity servers. Hence, all information stored in this process is prone to be breached. Besides, a strong security model is also lacking between Hadoop and HDFS. The correspondence among DataNodes and among the end users and DataNodes remains encoded. It has no validation of clients or administration. Even after Yahoo concentrated on including authentications in 2009, Hadoop still had constrained approved abilities (Yalla et al., 2016). In 2013, the Apache Software Foundation defined venture Rhino to include security highlights (Yalla et al., 2016).

Hadoop has the facility of data management that is scalable, rich in features and cost-effective for the masses. It has been a data platform of storing secret information for many organizations. The data stored in slots is saved but once it is brought together and made accessible for organizations over the masses, new security challenges arise. Big data in a Hadoop contains sensitive information related to financial data, corporate information, personal information and many such confidential data of clients, customers and employees. Hence, optimum protection of such data and ensuring that it remains free from any encroachment or tampering is of utmost significance (Rerzi, Terzi & Sagiroglu, 2015; Mehmood, Natgunanathan & Xiang, 2016; Bardi et al., 2014; Scrinivasan & Revthy, 2018; Derbeko et al., 2016; Gupta, Patwa & Sandhu, 2018).

Big data: hadoop security solutions

To elucidate the mentioned problems, a few activities have been appended to Hadoop to keep up with the equivalent (Vormetric Data Security, 2016; Jam, Akbari & Khanli, 2014):

Perimeter Security: Network Security firewalls, Apache Knox gateway

Authentication: Kerberos

Authorization: E.g. HDFS permissions, HDFS ACL3s, MR ACLs

Data Protection: Encryption at rest and encryption in motion. To provide security for data in HDFS, few available mechanisms are:

Authentication

Authentication implies user’s identification. Authenticators are answerable for gathering testimonials by the API (Application Programming Interface) consumers, authenticating them and publicizing the success or failure status to the clients or chain providers. Because of this primary check, uncertain users won’t be able to access the cluster network and trusted network. Identification is regulated by the client host. For strong authentication, Hadoop uses Kerberos (Vormetric Data Security, 2016; Jam, Akbari & Khanli, 2014), and LDAP (Lightweight Directory Access), AD (Active Directory) integrated with Kerberos, establishing a single point of truth.

Kerberos is a computer grid authentication protocol which generates “tickets” to allow the nodes communicating over an unprotected network to prove their identity to one another. The reliable server authentication key is placed in each node of the array to achieve authenticity of the Hadoop cluster node communication which will develop the HDFS array. It can effectively prevent non-trusted machines posing as internal nodes registered to the NameNode and then process data on HDFS. These components are used throughout the cluster. Hence, from the storage point of view, the legitimacy of the nodes in HDFS cluster could be guaranteed by Kerberos. It is completely entrusted by Hadoop for authentication between the client and server. Hadoop 1.0.0 version includes the Kerberos mechanism. Client requests an encrypted token of the authentication agent. A particular service can be requested from the server by using this. Password guessing attacks remains inoperative in Kerberos and thus multipart authentication is not provided (Zettaset, 2014).

Authorization

Authorization or restrict access is the method of securing the access within the data by the users as per the corporate policies or service provider. Authorization provider may also use an ACL (Access Control List) based authorization access called the Knox gateway (Vormetric Data Security, 2016; Jam, Akbari & Khanli, 2014) which is based on the evaluation of rules that comprises username, groups and IP (Internet Protocol) addresses. The aim of Hadoop’s developer is to design an authorization plan for the Hadoop platform to manage the authorization structure for Hadoop components.

Data protection

Data protection is a process to protect the data at rest or store and during transmission with the help of encryption and masking (Vormetric Data Security, 2016; Jam, Akbari & Khanli, 2014). Encryption is a technique which acts as an added layer in security in which data is encrypted (unreadable) during transmission or at rest. Hadoop employs the existing capabilities of data protection by providing the solution for data at rest and data discovery and masking. However, Hadoop’s security still needs some improvement. The work that has already been done by the researchers and practitioners on Hadoop is highly commendable. Several research studies have focussed on techniques to improvie the security of the data at rest as well as during transmission. Some of the relevant approaches have been discussed below:

Achieving secure, scalable and fine-grained data access control.

The work combines techniques of Attribute-Based Encryption (ABE), proxy re-encryption, and lazy re-encryption (Yu et al., 2010b). This integrated method accomplishes fine grainedness, scalability, and data confidentiality during data access control in cloud computing. In this work, data files are encrypted using symmetric DEKs (symmetric data encryption key of a data files) and later, encrypted DEKs with KP-ABE (public key cryptography primitive for one-to-many communications). Such a dual encryption technique is called hybrid encryption.

The KP-ABE technique is used for basic fuctions like the creation or deletion of files and user allocation with fine-grained data access control mechanism. User allocation is a big issue in this process and to achieve this, the author has combined proxy re-encryption with KP-ABE and distributed some tedious computational tasks to cloud servers. The cloud server stores secret key components and one dummy attribute corresponding to each user. When data owner does some modifications in the set of attributes while user allocation, the proxy re-encryption keys are generated and transferred to cloud servers. Later, cloud servers update their secret key on the basis of new re-encryption keys and re-encrypt the data files accordingly. Due to this, data owner is free from computation load during user allocation and do not need to stay online, since the cloud servers have already taken over this task after having the pre keys. Moreover, the burden of secret key updating and re-encryption of data file tasks are merged as single task using lazy re-encryption technique to save computation overhead of cloud servers during user revocation.

Secure data sharing by using certificate-less proxy re-encryption scheme.

This study stated that by using a public cloud, data can be shared securely. The research work presented a concept wherein a Certificate-Less Proxy Re-Encryption scheme (CL-PRE) is introduced (Xu, Wu & Zhang, 2012). According to this concept, an identity based public key is added to the proxy re-encryption technique. This removes the traditional identity problem of key escrow. This scheme requires no certificates for the authenticity of the public key. This scheme (CL-PRE) is used to decrease the figuring and correspondence cost for information proprietor.

Fully homomorphic encryption.

This research (Jam, Akbari & Khanli, 2014) proposed a design of trusted file system by combining the authentication agent technology with the cryptography fully homomorphic encryption technology.This is used for Hadoop which provides reliability and security from data, hardware, users and operations. This enables the user to prevent data breach along with enhanced efficiency of the application which is possible due to the encrypted data in the homomorphic encryption technology. Authentic agent technology also provides a range of techniques which are an integration of different mechanisms such as privilege separation and security audit that provides security of data in Hadoop system.

Fully homomorphic encryption technique gives the ability to various users to carry out any operation on encrypted data with same results, provided the nature of the data remains same, i.e., encrypted form throughout the operation. The data remains in encrypted form when processed with map reduce technique and stored safely in HDFS.

A novel data encryption in HDFS.

A new method for encrypting a file while uploading in HDFS has been proposed in this research work (Nguyen et al., 2013). The upload process is done along with the encryption process before uploading data on HDFS. In this method (Nguyen et al., 2013), the user selects a file to upload and provides a unique secret key for encryption of selected file. In this approach, user can feel the same experience when uploading a normal (without encryption) file to HDFS since the encryption is done in a fair manner. Also, this method utilises the characteristics of read/write operation to reduce the total time in HDFS. As an experiment, the author applied this technique on 32 MB file and observed that the encrypting upload and decrypting download process is usually 1.3 to 1.4 times faster than the conventional method. The major drawback of this approach is the key management because the keys are increased with respect to the users and to deal with them is quite challenging. Additionally, encrypting file sharing issue is also not possible with this approach. This proposed approach is lagging due to these two major issues and needs the dedicated attention of researchers and practitioners.

Secure hadoop with encrypted HDFS using AES Encryption/Decryption.

Security in Hadoop architecture is proposed in this paper by applying encryption and decryption techniques in HDFS (Park & Lee, 2013). In Hadoop, it is achieved by adding AES encrypt/decrypt function to Compression Codec. Experiments on Hadoop proved that the computation overhead is reduced by less than 7% when representative MapReduce job is done on encrypted HDFS.

Triple encryption scheme for hadoop-based data.

Cloud computing has the distinctive ability to provide users with customised, adaptable and trustworthy services at feasible costs. Hence, more and more users are availing of cloud computing. Given the rising demand of cloud appications, the protection of the cloud storage of data has become imperative. A method called novel triple encryption has been introduced in this paper (Yang, Lin & Liu, 2013) to achieve data protection at cloud storage. The Triple Encryption approach uses DEA (Data Encryption Algorithm) for HDFS files encryption, RSA for the data key encryption and finally IDEA for encrypting the user’s RSA private key. In this approach, DES and RSA based hybrid encryption technique is used to encrypt HDFS files and IDEA (International Data Encryption Algorithm) to encrypt the RSA based user key. In Hybrid encryption, DES algorithm is used to encrypt the files and get the Data key. This Data key is again encrypted by using RSA algorithm to provide additional security. The Data key can be decrypted by using private key only, therefore, it is always with the user. This method uses both symmetrical and asymmetrical encryption techniques, so called as hybrid encryption. This approach is tested and implemented in Hadoop based cloud data storage.

Attribute-group based data security access.

Due to various security issues, development and use of cloud storage has been decreased. To gain the confidence of user, author has defined an Attribute Group based data security access scheme to protect the data during network and data sharing features in cloud storage services. In this scheme (Zhou & Wen, 2014), the data owner has limited user rights and re-encryption on the data node reduces the computational cost along with the management of the clients. It also reduces the complexity of property and rights management. Also, the author uses cipher text CP-ABE encryption algorithm to secure the data at cloud storage. The centralised management of key distribution and Name Node based CP-ABE algorithms have advantages like more transparency for the user and easy managemnt of the user key as compared to the data owner key distribution technique.

Towards a trusted HDFS storage platform.

The mechanism for the protection of a Hadoop infrastructure has been explained in this research (Cohen & Acharya, 2014) to deal with the concept of creating a reliable HDFS and safety hazards. Also, the researchers of this paper figure out the relation between security mechanisms and their effect on its performance (Cohen & Acharya, 2014). In the discussion, the authors implemented trusted computing concepts on a Hadoop considering a threat based environment. This framework is based on the Trusted Platform Module (TPG) and implemented into a base environment. Furthermore, the authors have utilized hardware key protections encryption scheme for Hadoop and AES-NI for accelerating the encryption and compared results after their implementation. In addition, the authors have claimed that there is 16% of the overhead reduction on encryption and 11% overhead reduction while decryption during experiment on simulated 128 MB block data with the AES-NI instructions. This approach provides a concrete layered security design in Hadoop.

Security framework in G-Hadoop.

An approach has been introduced where Hadoop’s MapReduce task runs simultaneously on multiple clusters in a Grid environment called G- Hadoop (Jam, Akbari & Khanli, 2014; Zhao et al., 2014). G-Hadoop reuses user authentication and job submission mechanism of Hadoop in a Grid environment. But initially, Hadoop’s user authentication and job submission mechanism have been designed for a single cluster in a non-Grid environment. Therefore, G-Hadoop is an extended version of Hadoop MapReduce task. It established a secure link for user and target cluster by following the secure shell protocol. A single dedicated connection is allotted to each participating user in a cluster and each user has to log on to only those clusters for which they are authenticated. Unlike Hadoop, they have to design a new security framework for G-Hadoop with various security solutions like public key cryptography and Globus Security Infrastructure.

Concepts of proxy identity, user interface, and user instance, are embedded in this security framework to give better functions in a Grid environment (Jam, Akbari & Khanli, 2014; Zhao et al., 2014). This security framework introduced a single-sign on approach during user authentication and job submission process of the G-Hadoop. Also, this security approach protects the G-Hadoop system during threat environment, i.e., traditional attacks, abusing and misusing. The model of security framework is based on Globus Security Infrastructure (GSI). The utilization of SSL protocol for communication between the master node and the CA (Certification Authority) is also the key factor in security. GSI is based on single sign on process and uses asymmetric cryptography to provide a secure communication. GSI is a standard grid security which adapts to various techniques to provide necessary requirements in a Grid environment.This includes authentication, integrity of the messages and delegation of the authority from one entity to another in a grid environment. The user can only log-in into the master node after providing his authentication in the form of user name and password and submit jobs to the cluster. SSL handshaking is used in the security framework to establish a secure connection between DataNode and a NameNode.

Elliptic curve cryptography based security scheme for hadoop distributed file system.

This paper (Jeong & Kim, 2015) introduces a token based authentication scheme to protect HDFS stored data from security threats like breach and impersonation attacks. In this scheme, HDFS client authentication is done by the Data Node through block access token and functions as an extra layer of security along with the existing security, i.e., symmetric key HDFS authentication. Also, ECC encryption method is used for for authentication of anonymous keys and provides protection against external threats like security breaches or accidental exposures. This scheme adopts the hash chain of keys approach instead of a public key exchange approach which is a very common HDFS authentication protocol. Apart from providing protection to the sensitive HDFS data, it also improves the performance as compared to the public key-based HDFS authentication protocols.

Secure multi sharing in big data storage.

A method of privacy preserving security by using different mechanisms, i.e., anonymity, multiple receiver and conditional sharing is explained in this paper (Maheswari, Revathy & Tamilarasi, 2016). In this approach, to get the maximum security, Advanced Encryption Standard (AES) with Message Digest (MD5) & Data Encryption Standard (DES) have been employed to encrypt the data and authentication of data has been done using the DSA. Also, security and privacy preserving approaches have been used for the big data processing in the proposed framework. In this approach, owner uploads the data in cloud storage and after encryption, data is stored in HDFS. Thereafter, the data is shared among the multiple receivers. Cipher text is used to hide the identity of the sender and receiver whereas Anonymization mechanism is used to hide information of a particular receiver. A mechanism based on user and their received data category called conditional sharing starts working after receiving the receiver’s details. And, if the user’s category is matched with receiver’s data category, then the receiver gets authenticated and the transmission is started. Once the conditional sharing is complete, receiver retrieves the cipher text. The big data is shared with the cloud only if the result is secured. This proposed algorithm is verified for small data sets only.

Towards best data security.

In this paper, the author has described about the enormous information and its safety issues (Tian, 2017). Also, he has described about the existing ways to improve the security of enormous information like security hardening methodology with attributes relation graph, attribute selection methodology, content based access control model and a scalable multidimensional anonymization approach. The author of this paper (Tian, 2017) has proposed an intelligent security approach based on real time data collection and threat analytics to detect the threat before the security breach takes place.

HDFS data encryption based on ARIA algorithm.

In this paper (Song et al., 2017), the author has presented an encryption scheme based on South Korea’s ARIA encryption scheme to protect the HDFS data in Hadoop. The ARIA algorithm uses 128-bit block for data encryption. In this approach, variable length data (not necessarily the 128-bit data) is divided into HDFS blocks. The proposed ARIA based algorithm provides the same level of data security at cost of only 23% performance degradation (during query processing) compared to AES algorithm. In addition, the researchers explained the future of ARIA based encryption scheme in genuine word applications like area based administrations and financial related data handling.

Chaos-based simultaneous compression and encryption for hadoop.

This paper (Usama & Zakari, 2017) introduced a framework based on a masking pseudorandom key stream to increase the encryption quality and provide robust encryption security & compression during read and write operation when integrated in HDFS. Also, the researchers have proposed a scheme for Hadoop using simultaneous compression and encryption to solve the implementation issues. The enhancement consequently improves the speed and efficiency of the algorithm. The proposed algorithm is highly compatible with Hadoop and provides efficient encryption security and compression during storage of data. Various experimental results concluded that the performance of the cluster in Hadoop gets reduced when compression and encryption operations are done separately because they need a significant volume of data for both the operations. This proposed algorithm can compress and encrypt the data simultaneously during MapReduce which reduces the required data space with minimum network resources. The proposed algorithm has passed edits security analysis test with a 99% confidence interval. Further, all NIST SP800-22 assays are successfully passed on cipher text generated from the plaintext.

Data encryption based on AES and OTP.

This research paper (Mahmoud, Hegazy & Khafagy, 2018) has highlighted a method to improve the upload and download time with reduction of encrypted file size by AES and OTP algorithms in HDFS. The authors performed encryption and decryption by two different ways which are based on AES and AES-OTP algorithms. The researchers chose cipher block chaining with the ECB mode of AES algorithm for handling HDFS blocks and OTP algorithm is used as a stream cipher. This keeps length of the plaintext same. For decryption, a private key is required which is always in the custody of user. In this method, when client has mentioned to transfer a record to HDFS, the application server creates an arbitrary key which is then separated into 2 keys for doing multi encoding and unscrambling by utilizing AES-OTP algorithm.Moreover, the authors have compared the file encryption time among Generic HDFS, encrypted HDFS by AES and HDFS encrypted file by AES with OTP. The results show that the AES with OTP algorithm increased the encrypted file size by 20% of the original file. The researchers also executed parallel decryption processing in Map Task to improve performance.

Two-layer selective encryption using key derivation method.

In this paper (Vimercati et al., 2007), the authors have explained the use of two-layer selective encryption technique based on key derivation method to implement the authorization policy (Atallah, Frikken & Blanton, 2005). In this method, the user assigns a secret key corresponding to each file which is encrypted using a symmetric key. The owner creates public tokens by using his secret key to allow any user further. Later, these public tokens along with token distribution task are transferred to the semi-trusted server. To derive the decryption key for a file, a minimal number of secret key per user and a minimal number of encryption key are required since the server cannot derive decryption key of any file with the available public tokens. The file creation and user grant/revocation operation gets complex as the number of users increases. This makes the suggested method unscalable (Yu et al., 2010a). Also, the user access privilege accountability is not supported in this method.

Security and privacy aspects in mapreduce on clouds.

Hadoop uses the filters in Vigiles (Ulusoy et al., 2014) for a fine grained access control framework. These filters are coded in Java by security administrators and handled authorization by means of per-user assignment lists. On the other hand, in GuardMR, filters are allocated with limited roles on the basis of subject and a formal specification approach for the definition of filters is proposed. GuardMR and Vigiles rely on platform specific features for regulating the execution of a MapReduce task such as the Hadoop APIs and the Hadoop control flow and do not need the Hadoop source code customization. Vigiles and GuardMR have observed apractically low implementation overhead which means that they do not provide any support for context aware access control policies (Colombo & Ferrari, 2018). In (Derbeko et al., 2016) authors considered security and privacy challenges and urgent requirements in the scope of MapReduce and reviewed the existing security and privacy protocols for MapReduce including AccountableMR and TrustMR. The study also provides a comparison of several security algorithms, protocols and frameworks for MapReduce framework.

Hybrid storage architecture and efficient mapreduce processing for unstructured data.

In this paper (Lu et al., 2017), a technique called Hybrid Storage Architecture is proposed. With this technique, different kinds of data stores are integrated to the model and it also enables the strorage and process of the unstructured data.To execute MapReduce-based batch-processing jobs, various partitioning techniques are applied which are based on the said technique. The paper also demonstrates the utilization of the characteristics of different data stores for building a smart and an efficient system. The partitioning techniques leverages the unified storage system thus reducing the I/O cost and improves the large-scale data processing efficiency marginally.

Towards privacy for mapreduce on hybrid clouds using information dispersal algorithm.

In Cheikh, Abbes & Fedak (2014), to ensure privacy for MR in a hybrid computing environment based on the Grid’5000 platform, an algorithm known as information dispersal algorithm is required which comprises both untrusted infrastructures (such as, desktop grids and public clouds) and trusted infrastructures (such as, private clouds).

SEMROD: secure and efficient mapreduce over hybrid clouds.

SEMROD (Oktay et al., 2015) firstly segregate the data into sensitive and non-sensitive data groups and then send the non-sensitive data to public clouds. Private and public clouds execute the map phase. However, the private cloud pulls all the outputs includng outputs of the map phase containing sensitive keys. Also, it executes the reduce phase operation only on record associated with sensitive keys andignores the non-sensitive keys. On the other hand, a public cloud execute the reduce phase on all outputs without knowing the sensitive keys. Finally, a sensitive key is generated by removing the duplicate entries with the help of filtering step.

MtMR: ensuring mapreduce computation integrity with merkle tree-based verification.

Proposed MtMR (Wang et al., 2015) is a method based on Merkle tree based verification to ensure the high integrity of the MapReduce tasks. It performs two rounds of Merkle tree based verification for the pre-reduction and restoration phases and covers MapReduce in a hybrid cloud environment. In each round, MtMR samples a small portion of reduces task input/output records on the private cloud and then applies the Merkle tree-based verification. The authors believe that MtMr can significantly improve the results while producing moderate performance overhead.

Security threats to hadoop: data leakage attacks and investigation.

This article (Fu et al., 2017) presents an automatic analysis method to find any data leakage attacks in Hadoop. It also presents a forensic framework including an on-demand data collection method in which it collects data from the machines in the Hadoop cluster on the forensic server and then analyzes the same. It can detect suspicious data leakage behaviors and give warnings and evidence to users using its automatic detection algorithm. And, collected evidences can help to find out the attackers and reconstruct the attack scenarios. The authors of the paper have also talked about the security concerns of HDFS (or Hadoop) and presented some possible data leakage attacks in it.

VC3 and M2R in mapreduce computation.

VC3 (Schuster et al., 2015) uses SGX to achieve confidentiality and integrity as part of the MapReduce programming model and requires a trusted hardware to perform computation. VC3 is not allowed to perform system calls but works and follows the executor interface of Hadoop. On the other hand, M2R (Dinh et al., 2015) offer mechanisms for dropping network traffic analysis leakage for MapReduce jobs.

Preserving secret shared computations using mapreduce.

The main reason of cloud insecurity is the loss of control over the data which can cause serious harm to the confidentiality of customer using cloud-computing. This problem can be overcome by providing secure computing environment and data storage (Sudhakar, Farquad & Narshimha, 2019). Also, techniques like encrypted representation and secret sharing techniques have emerged that offer verified security along with relatively efficient processing but are limited to only computing selection queries (Dolev et al., 2018).

Privacy preservation techniques in big data analytics.

In this paper (Mohan Rao, Krishna & Kumar, 2018), authors have described about the various privacy threats and preservation techniques and models along with their limitations. The authors also proposed a Data lake based method for privacy preservation of unstructured data. Data lake is a repository to store raw format of the data either structured or unstructured, coming from different sources. Apache Flume is used for data ingestion from different sources and for their processing; data is transformed to HIVE tables. Also, Hadoop MapReduce using machine learning or vertically distributed can be applied to classify sensitive attributes of data whereas tokenization is used to map the vertically distributed data.

Major findings from the literature

After a cautious and focused study of various methodologies/approaches on big data and Hadoop security, the following observations have been made:

• Hadoop stores the data on multiple nodes in a distributed manner while metadata and edit logs are stored on name nodes. The communication of data happens between client node and data node. Hence, multiple nodes are used in the Hadoop framework. The data is vulnerable to the hacks during the transmission, as it is not encrypted by default. Various communication protocols are used for internode communication. The available approaches or solutions for securing the data transmission include Kerberos, Simple Authentication and Security Layer (SASL), etc. However, these traditional approaches are not effective and sufficient enough to secure big data.

• Data that is stored in fixed storage is known as data at rest or at storage level. Initially the stored data is prone to security attacks being not encrypted. Since, Hadoop works on the principal of spreading data across multiple nodes, consequently it is exposed to all insecure entry points. There are numerous methods available for data encryption in Hadoop. As Hadoop deals with large volume of data, it takes time in the encryption/decryption process. In order to maintain the performance, it is important to use an encryption technique that is fast enough to encrypt/decrypt. According to the studies, the encrypted data increases in the size almost by one and half time of the original data so the file upload time also gets affected.

• Cloud providers need to design a cost-effective infrastructure that understands customers’ needs at all levels. To meet the requirements, it is needed to share the storage devices amongst the multiple users, which is known as multi-tendency. But sharing of resources results in security vulnerability. If proper security measures are not implemented, then the attacker is able to get easy access to the customer’s data, more so in the case of using the same physical device.

• Companies would never know if the data is being used by someone else or not, because they don’t have direct control over their data. The lack of resource monitoring mechanisms creates many security issues.

• Customers have to rely upon trust mechanism as an alternate security measure in which they have to control data and their resources. Cloud providers also provide certificates of operations of a secure provider to the customers. The certificates are well authenticated with established standards.

• The security capabilities which are for “non big data” are needed for big data also to ensure client verification, management of data masking and encryption.

Materials & Methods

Data Encryption based on Attribute Based Honey Encryption (ABHE)

In Hadoop, the inherent security feature is simple file permission and access control mechanisms. In such context, encryption is the best technology applied for securing HDFS files that are stored in DataNodes. Further, while processing MapReduce transferring files among DataNodes, encryption is the best solution. We can use cryptography for data protection in Hadoop, solution to data confidentiality and data integrity can be achieved using encryption technique. Cryptography keys can be categorised into: secret key cryptography and public key cryptography. Public key is known as asymmetric key cryptography (Dyer et al., 2013) while secret key is symmetric secret key cryptography which is used in stream ciphers for generation of password based encryption (Vinayak & Nahala, 2015).

Encryption is mainly used to ensure secrecy. Encryption actually means secret writing which was initially used by ancient humans desiring to store secrets. In the past, encryption was available only to Generals and Emperors, but today it is used by nearly everyone, every day, every time whenever a credit card transaction, data storage and node to node communication is done, phone call is made, secure website is used; encryption techniques are used. Efficacy of an encryption algorithm depends on the key length (Ebrahim, Khan & Khalid, 2013). However, the available encryption algorithms are considered to be secure. But depending on the time and computing power, they are also susceptible to intrusions (Yin, lska & Zhou, 2017). The present encryption techniques are also beset with vulnerabilities, for instance, when decrypting with a wrongly guessed key, they yield an invalid looking plaintext message, while decrypting with the right key, they give a valid-looking plaintext message, confirming that the cipher-text message is correctly decrypted (Yin, lska & Zhou, 2017).

In the same row, the honey encryption has been proposed by Jules and Ristenpart (Juels & Ristenpart, 2014). It is a concept which addresses vulnerability discussed in the previous paragraph and makes the password based encryption (PBE) more difficult to be broken by brute-force. Traditional encryption methods would show random text with no meaning at all when decrypting is done with wrong key and hence confirming its invalidity. On the contrary, honey encryption shows plausible looking text even when the key is wrong so the attacker won’t know if the guessed key is the right one. This unique approach slows down the attacker by fooling him and increases the complexity of password guessing as well as cracking process. There are few other technologies that share same term “honey”. For example, Honeywords (Juels & Ristenpart, 2014) are a password that are used as decoy and generates an alert and notifies the administrator if used. Honeypots (Owezarski, 2015), Honeynet (Kim & Kim, 2012), and Honeyfarm (Jain & Sardana, 2012) are some other examples of luring systems. Honey encryption is related to Format-Preserving-Encryption (FPE) (Bellare et al., 2009) and Format-Transforming-Encryption (FTE) (Dyer et al., 2013). In the FPE, both the plaintext and cipher-text message are same whereas it is not the case in FTE.

In Honey Encryption, the messages are stored to a seed range in the seed space. Seed space and message space are different, so the cipher-text message space is different from the message space. Vinayak & Nahala (2015) used the HE scheme in MANETs to secure Ad-hoc networks against brute force attacks. Tyagi et al. (2015) applied HE technique to protect simplified text messages and credit card number that are susceptible to brute force attacks. Choi, Nam & Hur (2017) proposed schemes to solve human typo problems with message recovery security. Legitimate user may get confused seeing the different result than expected if there was some mistake in typing the password correctly. Edwin Mok et al. (Tan & Samsudin, 2018) came up with an eXtended Honey Encryption (XHE) by adding additional security measures on the encrypted data. However, Honey Encryption is still difficult to be applied in certain applications. For example, if the attacker has some clue about the data which is encrypted, suppose he has a part of the original data, he can easily tell which result is bogus and which is the correct data by matching the data with the decrypted result. However, it is possible to brute force honey encryption if the attacker has crib that must match with it to confirm its legitimacy (Wikipedia, 2019). It is still vulnerable and susceptible and further researches are going on. To overcome its limitations it must be expanded further by bringing out new security methods.

This persuades the authors of this paper to develop an in-depth understanding of data security and privacy to solve issues related to Honey encryption. This paper aims to focus on fixing the vulnerabilities in Honey encryption and making it more secure. The authors have designed and implemented the attributed based Honey encryption as an extension of the public key encryption. This would enable the users to encrypt and decrypt messages based on users’ attributes. Only if the user matches the predefined attributes will the user be able to decrypt the message. It will help to keep the attacker away by blacklisting them.

Proposed encryption algorithm

The proposed encryption algorithm is a more secure version of honey encryption. The encryption algorithm provides two tier securities so that it can overcome the limitations prevailing in existing encryption techniques. The proposed algorithm is termed as Attribute-Based Honey Encryption (ABHE). Its 128/256 bits encryption algorithm will perform two layers of encryption in order to enhance security and effectiveness. The use of Cipher text Policy- Attribute Based Encryption (CP-ABE) (Zhao, Wei & Zhang, 2016; Varsha & Suryateja, 2014; Shobha & Nickolas, 2017) has been proposed in the algorithm. In the algorithm user’s private-key is superimposed with an access policy over a set of attributes within the system. A user can decrypt a cipher text only if his attributes satisfy the set of rules of the respective cipher-text.

Firstly, a set of attributes are chosen from the file to be encrypted; then a set of rules/policies are created for these attributes. On the basis of these rules, the given file is encrypted. Further, for more security the encrypted file is again protected by password. As this password is based on honey encryption, it creates a set of honey words. The encrypted file is passed on and may be received by different users. Now according to the proposed algorithm, only the user having the desired set of attributes or the password would be able to decrypt the data. If someone wants to decrypt the encrypted file, he/she will have to enter the correct password. If password does not match, the user will be treated as intruder and previously set honey words will be displayed to him. If the password matches, the genuine user has to enter the private key which has been already created while encrypting the file. Again, if private key does not match, the user will not be allowed to access the file. On matching, the user will be able to successfully decrypt the file. The overall process will provide better security for files.

ABHE algorithm for data security

Input:	Plain Text file	
Output:	Encrypted file 	
Step 01:	Generate Private Key 	
Step 01.a:	Set of attributes is specified that describe the key.	
Step 01.b:	Output private key ‘q’	
Step 02:	Encryption:(The algorithm encrypts File ‘F’ with policy ‘P’ and outputs the cipher-text)	
Step 02.a:	Selects the file to be encrypted and set of attributes.	
Step 02.b:	Encrypt a file F using a set of attributes occurring in the policy ‘P’	
Step 03.c:	Generate cipher-text CT	
Step03:	Encrypted file (in step-02) is protected again by the password	
Step 04:	Generate honey words and present it to user.	
Step 05:	Decryption: (Decryption algorithm gets as input an encrypted file which is protected by the password. Cipher-text CT is produced by the encrypted algorithm, an access policy ‘P’ under which CT was encrypted.)	
Step 05.a:	Input is encrypted file	
Step 05.b:	Enter the password; if password matches, the cipher text CT is decrypted, otherwise intruder is detected.	
Step 05.c:	User applies y number of attributes to compute private key	
Step 05.d:	If key matches, file is decrypted and output the corresponding original file ‘F’, otherwise it outputs NULL.	

Authors have introduced a method to enhance the security level in which the data encryption and key management server are put together and provided a unique key for each application or cluster with HDFS encryption. When HDFS encrypted file entering into the Apache Hadoop environment, it remains in encrypted form at storage after processing. The results including intermediate results are always stored in encrypted form within the cluster in a file system having non HDFS form. At client level, data has been divided into smaller chunks by using parallel computing technique and stored at HDFS in encrypted form. Also, the Map Reduce tasks can be done on encrypted data directly and decrypt before processing after importing the corresponding decryption library. Input to a MapReduce job is divided into fixed-size pieces. These are the input splits which is a chunk of the input that is consumed by a single map. At Map function, the input data is processed in decrypted form and stored output data in encrypted form into the HDFS. The Reduce function is executed on the intermediate results of Map function after decryption and the stored final output data in again encrypted form into the HDFS and provided access to the authorized clients only. Decryption process is replica of encryption process and both these methods are simple, cost-effective and scalable without deteriorating the performance, scalability or functionality. So, they are easy to recommend and effectively address the security deficiencies with big data clusters.

Evaluation of performance of the proposed algorithm has also been done. The performance parameter includes encryption–decryption time (rate of encryption is given by encryption time and rate of decryption is given by decryption time), throughput of encryption-decryption where throughput is calculated as the size of plain text (in MB or GB) is divided by the total time during encryption-decryption. The speed and power conumption of encryption-decryption process are mainly dependent on the throughput of the encryption-decryption scheme, as it defines the speed of encryption-decryption. In case of encryption-decryption, as the throughput increases, power consumption decreases. Also, authors have compared results with the exsisting HDFS encryption algorithms namely AES and AES-OTP with different file sizes (varies from MB to GB). The performance parameters results have shown that the proposed ABHE scheme with Hadoop environment is a considerable improvement over AES, AES with OTP (Integrating with Hadoop). Also, the proposed algorithm provides the security for data stored at HDFS and distributed computing against all side channel attacks, brute force attacks and collusion attacks. Detailed description is given in the next section.

Results

Implementation

AES is propositioned to be better than the other secure approaches that address the secure data processing using Hadoop HDFS and MapReduce job in context of data encryption. To support this claim, the performance of proposed ABHE algorithm has been evaluated in this section and performance of the proposed ABHE algorithm has been compared with the existing algorithms, i.e., AES and AES-OTP, while doing the same experiment in a standard Hadoop setup. The performance is evaluated in terms of throughput and power ponsumption by doing the encryption-decryption techniques on different sizes of files (size varies from MB to GB).

Implementation environment

The implementations and experiments are based on Hadoop cluster. The Hadoop cluster consists of one host which runs on laptop with Intel core i3-2330M processor 2.20 GHz with Turbo Boost upto 2.93 GHz and 4GB RAM. In this, one of the host is tagged as NameNode and other is used as a DataNode. NameNode playsa role of centre point in cluster and all information about the stored data is saved on it. For simplicity, there is only one NameNode and one DataNode in a Hadoop cluster with one run. DataNode provides the physical space for storing data. The operating system of the host is Linux with CentOS 6.4, Ubuntu-14.04 (GNU/ Linux 3.13.0-24-generic x86-64). On top of the operating system is Hadoop with version 2.7.3. Single node architecture of Hadoop is used in which all components of the Hadoop are on the same node. Implementation and stand-alone applications are written in Java. As Hadoop requires JDK for its working so, Java Open Java Development Kit (JDK) is installed on the system using the <apt-get>command. The running component of the Hadoop can be checked using the <jps>command. HDFS distribution process overcomes outages, acts as a backup while maximising the availability of the services.

Results of the experiment

In this section, we present the results and analysis of our proposed algorithm versus the available securing approaches. In the proposed encryption technique, at first, we apply the attribute based encryption which is based on cipher text policy based attribute encryption. The proposed approach uses a specific type of encrypted access control wherein user’s private-key is super imposed with an access policy over a set of attributes within the system. A user can decrypt a cipher text only if his attributes satisfy the set of rules of the respective cipher-text. An enhance security to ensure full safety against all side channel & brute force attack, the proposed algorithm is combined with Honey encryption algorithm. The combination of these two algorithms, i.e., the ABHE provides a stronger security against confidentially & brute force attack and all side channel as well as collusion attacks as encryption is not easy to break and get the actual data.

The performance of ABHE has been calculated in terms of file size, encryption time, decryption time and power consumption and compared with two existing encryption algorithms namely; AES and AES with OTP applying on different sizes of text files. The working of the proposed algorithm has been demonstrated in Figs. 1 and 2.

File size

The proposed encryption algorithm reduces the encryption-decryption time without affecting the size of original file. Here, the file named 2048MB.txt is of size 1.12 GB and contains 9 blocks starting from 0 to 8. The size of block 0 to block 7 is remains unchanged after encryption while size of block 8 is changed from 125449781 bytes to 125449792 byte which is insignificant and shown in Figs. 3–6. Also, the output file size when encrypted a file of size 1 GB using AES, AES-OTP and the proposed approach is compared which is shown in Table 1.

Figure 1 Working of the proposed ABHE encryption algorithm when attribute and password are entered for 64 MB file size (for both encryption and decryption).

Figure 2 Working of proposed ABHE encryption algorithm when right and wrong password are entered for 64 MB file size.

Encryption time using ABHE

This is the time taken by encryption algorithm to produce a cipher-text from plain-text. Encryption is performed while writing the file on Hadoop so that the stored data can be saved from various attacks. This process involves a number of steps which has been shown as follows:

i. HDFS client interacts with NameNode by calling the create() function on Distributed file system (DFS).

ii. DFS sends a request to NameNode to create a new file.

iii. NameNodes provide address of the DataNode, i.e., Slave which is based on the availability of space and capacity in DataNode on which HDFS client is going to start writing encrypted data.

iv. The HDFS client starts entering the attributes to encrypt the file. After that for more security, it applies the password which is based on Honey encryption. Now the HDFS client starts writing data though FS Data OutputStream to specific slave for a specified block.

v. The slave starts copying the block to another slave when HDFS client has finished writing the blocks.

vi. During the block copying and replication process, metadata of the file is updated in the NameNode by Datanode. (DataNode provides the periodically heartbeat signal to the NameNode).

vii. After the successful completion of write operation,DataNode sends the acknowledgement to HDFS client through DFS.

viii. After that HDFS client closes the process.

ix. Write operation is closed after receiving the acknowledgement from HDFS client.

The complete operation (with above steps i.e., i, ii, iii…) is explained in Fig. 7. As shown in Table 1, it took 12.9751 min for the encrypted HDFS using AES algorithm, whereas it took 11.2511 min for the encrypted HDFS using AES with OTP algorithm. On the other hand, the proposed approach took only 6.08 min to encrypt 1GB file in HDFS as shown in Table 2.

Figure 3 Size Block one before encryption.

Figure 4 Size of Block one after Encryption.

Figure 5 Size of Block eight before encryption.

Figure 6 Size of Block eight after encryption.

Data Decryption Time using ABHE

It is the time taken by decryption approach ABHE to produce the plain-text from cipher-text. With our proposed cryptographic scheme, whenever a node will try to read a file on HDFS it will first have to decrypt the file. Then only it will be allowed to perform reads operation. This has been done in the proposed approach to filter out the intruders or unauthorized access. Following is the step-by-step process on how Read operation is performed in HDFS with the proposed approach:

i. First of all HDFS client interacts with NameNode by calling the read function on Distributed File System (DFS).

ii. DFS sends a request to NameNode for reading a file.

iii. NameNode provides address of the DataNode, i.e., Slave on which HDFS client will start reading the data.

iv. For HDFS client to start reading data through FS Data InputStream from specified slave and from a specified block, firstly it has to enter the correct password. If password does not match, the user will be treated as an intruder. If the password matches, the genuine user has to enter the private key which has already been created while encrypting the file. Again, if private key does not match, the user will not be allowed to access the file. On matching, the user will be able to successfully decrypt the file.

v. After a successful completion of read operation, HDFS client terminates read operation.

vi. Read operation is closed after receiving the acknowledgement from HDFS client.

vii. As it has been shown in step 4, the proposed approach provides dual layer security to the data stored in HDFS. Step-wise demonstration of decryption operation is shown in Fig. 8.

When using the proposed approach for decryption of 1GB file on Mapper job, it took 6.73 min. On the other hand, the existing algorithms, i.e., AES and AES with OTP respectively to 14.0841 and 12.2115, respectively for decrypting 1 GB file as shown in Table 3.

Table 1 File size comparison among AES and AES with OTP algorithms and the proposed ABHE algorithm.

File size (MB)	AES algorithm(MB)	AES with OTP algorithm (MB)	Proposed ABHE algorithm(MB)	
64	96.0	74.7	64	
128	192.8	149.3	128	
256	384.0	298.7	256	
512	768.0	597.3	512	
1,024	1,536	1,228.3	1,024	

Figure 7 Writing a file with encryption in HDFS.

Table 2 File encryption performance comparison among AES and AES with OTP algorithms and the proposed ABHE algorithm.

File size (MB)	AES algorithm(Minutes)	AES with OTP algorithm (Minutes)	Proposed ABHE algorithm(Minutes)	
64	0.8704	0.7311	0.026	
128	1.8216	1.3820	0.168	
256	2.7396	2.5484	0.45	
512	6.6682	4.8780	1.35	
1,024	12.9751	11.2511	6.08	
Total Encryption Time	25.0749	20.7906	8.074	
Throughput of Encryption (MB/Minutes)	79.12	95.42	245.72	

Figure 8 Reading a File with Decryption in HDFS.

Table 3 File decryption performance comparison among AES and AES with OTP algorithm and the proposed ABHE algorithm.

File size (MB)	AES algorithm(Minutes)	AES with OTP algorithm (Minutes)	Proposed ABHE algorithm(Minutes)	
64	1.3056	1.0950	0.03065	
128	2.1859	1.6560	0.168	
256	2.8641	2.6554	0.45	
512	8.9494	6.5361	1.35	
1,024	14.0841	12.2115	6.73	
Total decryption time	29.3891	24.154	8.72865	
Throughput of decryption (MB/Minutes)	67.50	82.13	227.29	

The values for each criterion was logged and graphically plotted to represent the results as shown in Figs. 9 and 10. Further, these figures show the comparative time taken (in minutes) during the encryption and decryption process by different algorithms i.e., AES, AES with OTP and Proposed Algorithm (ABHE). From Figs. 9 and 10, it is clear that the proposed algorithm is taking less time for encryption and decryption as compared to other existing algorithms in Hadoop environment.

Figure 9 Encryption Time (minutes) of AES, AES with OTP and Proposed ABHE Algorithm.

Figure 10 Decryption Time (minutes) of AES, AES with OTP and Proposed ABHE Algorithm.

When the proposed ABHE algorithm is integrated with Hadoop, it showed better performance than the previously available cryptographic algorithm. From the results of Tables 1–3, it is clear that:

• The proposed algorithm ABHE is taking less time to encrypt and decrypt text files than the AES, AES with OTP algorithms.

• The throughput of ABHE is very high as compared to the AES, AES with OTP algorithms.

• As the throughput increases, the power consumption decreases, hence the power consumption of ABHE is low than that of the AES, AES with OTP.

Furthermore, for analyzing the performance of the proposed encryption technique with sharing data between two different DataNodes in Hadoop environment, the same has been simulated with random text file size 712 MB (in terms of block size) before and after encryption shown in Figs. 11 and 12. Also, the Browse directory show that encrypted file and abc.txt non-encrypted file HDFS in Fig. 13.

Figure 11 Size of Block five before Encryption.

Figure 12 Size of Block five after Encryption.

Figure 13 Browse directory show that encrypted file and abc.txt non-encrypted file HDFS.

Discussion

The proposed study has been able to successfully solve the weaknesses present in the security approaches available for big data security. The significance of the proposed work is as follows:

• Proposed encryption technique which uses the concept of Attributes Based Honey Encryption (ABHE) may help to securing sensitive information stored at HDFS in insecure environment such as the internet and cloud storages.

• Proposed technique provides both HDFS and Map Reduce computation in the Hadoop as well as cloud environment to secure and preserve the integrity of data during execution or at rest. Therefore, we have directed our efforts in securing the data transfer and computation paradigm in Hadoop environment by using chipper text policy attributes based honey encryption and Honey encryption for secret share of tuple of data and sent them to the cloud in a secure manner.

• The chipper text policy attributes based encryption makes the application secure and has a high performance when compared with the rest of the encryption techniques. Also, it provides the secure data transfer to all cloud applications.

• In the proposed algorithm, we have assured the data security by using simplified chipper text policy attribute based encryption with Honey encryption which is difficult to decrypt by any unauthorized access.

• The user authorization access is based on the user define policy which reflects the overall organizational structure and also, depends upon a set of attributes within the system.

• With the proposed algorithm, the security of data is not only dependent on the secrecy of encryption algorithm but also on the security of the key. This provides dual layer security for the data.

Conclusion

In this proposed approach, we mainly concentrated on protection of big data stored in HDFS by integrating the proposed ABHE algorithm with Hadoop key management server. In a nutshell, for ensuring data security in Hadoop environment through the proposed encryption technique, HDFS files are encrypted by using attribute based honey encryption through the proposed ABHE algorithm. For evaluating the suggested technique, we carried out some experiments using two data nodes. Our objective was to experiment and gauge the effectiveness of ABHE algorithm. For accuracy in sharing secret key, data sharing between different clients and speed with which each file stored in HDFS. As the proposed ABHE algorithm, execution time (a function of encryption time) is less as compared to the other available approaches. This proves that the proposed technique is fast enough to secure the data without adding delay. Also, the proposed ABHE algorithm has a higher throughput which proves its applicability on big data. It provides a feasible solution for secure communication between one DataNode to other DataNode. The proposed encryption technique does not increase the file size therefore it saves the memory and bandwidth, and hence reduces traffic in a network. Also, it has an ability to encrypt structured as well as unstructured data under a single platform. Only HDFS client can encrypt or decrypt data with accurate attributes and password. The Proposed technique provides a dual layer security for all DataNode as data is not confined to a specific device and clients can access the system and data from anywhere. This encryption approach may be reckoned as a premise for visualizing and designing even more robust approaches to ensure optimum security of big data.

Supplemental Information

Supplemental Information 1 Code

Click here for additional data file.

Additional Information and Declarations

Competing Interests

Author Contributions

Data Availability

The authors declare there are no competing interests.

Gayatri Kapil conceived and designed the experiments, performed the experiments, performed the computation work, prepared figures and/or tables, authored or reviewed drafts of the paper, and approved the final draft.

Alka Agrawal conceived and designed the experiments, performed the experiments, analyzed the data, authored or reviewed drafts of the paper, and approved the final draft.

Abdulaziz Attaallah conceived and designed the experiments, analyzed the data, prepared figures and/or tables, authored or reviewed drafts of the paper, and approved the final draft.

Abdullah Algarni performed the experiments, performed the computation work, authored or reviewed drafts of the paper, and approved the final draft.

Rajeev Kumar and Raees Ahmad Khan conceived and designed the experiments, performed the experiments, analyzed the data, prepared figures and/or tables, authored or reviewed drafts of the paper, and approved the final draft.

The following information was supplied regarding data availability:

Code is available as a Supplemental File.

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
