# Peer review of "Attribute based honey encryption algorithm for securing big data: Hadoop distributed file system perspective"

_PeerJ Computer Science, doi:10.7717/peerj-cs.259_

## Round 0.1 · original submission · Major Revisions

Please see whether you are able to address the comments of the reviewers, they are both skeptical concerning the possibility to revise the paper to an acceptable form. You may want to refer to the recent works of Shantanu Sharma.

[]

·

Basic reporting

- I would suggest sending the article to English-native person. There are many parts of the paper that are hard to understand and many sentences that do not make sense.
For example (examples only): 2 first sentences in Introduction are not clear and there are some "s"-s missing and Hadoop Security first paragraph is not clear at all.

- It would be better to concentrate on the most relevant topics of the paper. For instance, sections 2.3 and 2.4 are not really relevant to the paper and can be shortened considerably.

- Relevant work: I liked the summary of the related work, this is a great idea. The format is not correct, no need to make each paper into a separate sub-section, also some sub-sections should be shortened. In the summary fo the related work: point 2 is not convincing as even though the encryption/decryption take time due to its distributed nature Hadoop (HDFS) will suffer from smaller penalty.
Point #4 - not sure how this is a conclusion from the related works.
Point #5 ends abruptly and not clear what was the purpose of it.

- Section 3: should explain more about Honey encryption and also clearly define the addition made in the paper.

- HE is weaker than AES and was originally targeted to support authentications and not data at rest. It should be mentioned, as the speed up in encryption and improvement in the data size comes with a price.

- Section 3.2: the algorithm is not very clear and there are some "bugs" in definitions. For example (one out of ...): P is defined as private key in step 01.b and as policy in step 02.

Experimental design

It is not clear if the reported results are from a single run or not.
If not, then STD should be reported as well.

In context of HDFS, it is important to run on more than one node. Both due to data replication and etc.

Validity of the findings

As mentioned above HE is weaker than AES and was originally targeted to support authentications and not data at rest. It should be mentioned, as the speed up in encryption and improvement in the data size comes with a price.

The paper does not mention how the data is shared between different clients. This might be a problem with password and private key.
Another issue that is not mentioned is the data replication of HDFS, how are the keys managed and synchronized.

Another not clearly explained issue is how map-reduce paradigm (among others) are supported with the encryption and password scheme.

Reviewer 2 ·

Basic reporting

Not clear
Ambiguous
Non-professional article structure
No formal results

Experimental design

No rigorous investigation

Validity of the findings

Conclusions are not well stated

Additional comments

The paper provides a secure job execution approach for MapReduce based on public/private key and passwords. However, the approach is not practical as well as secure. For example, the proposed approach suggests that mappers/reducers at the cloud will decrypt the file and then execute the computation. At this time, the adversarial cloud can observe the computation and data. Decryption operation at a mapper/reducer cannot secure the data/computation. The authors should have to check Intel SGX and see: VC3: Trustworthy Data Analytics in the Cloud using SGX. Another problem is that the paper is not able to motivate the problem and providing any new contribution.

---

## Round 0.2 · Major Revisions

Please address the concerns of the reviwers

·

Basic reporting

The language of the paper could be improved. There are not many typos but there are unclear sentences.
Despite that, the paper is clear structured well.

Examples (not all of them) of misuse:
Line 71 - typo
line 80 - small letter
line 101 - "atmosphere"?

line 106 - B includes C (115) and D (125)

line 120 - "instrument" ?
Kerberos description is not clear and very high-level

line 138 - It is hard to say that Hadoop security is at early stage, as the first paper cited was in 2010 and since 2009 Yahoo is working on addition security mechanisms.
It could be a case that the existing protection is unsatisfactory, but then the reason for this should be listed.

line 394: "the technique" - what technique?
Also, the separation is not really of cryptography keys but rather of encryption methods.

Experimental design

The experiment results are somewhat unclear to me.
Specifically the source of the performance improvement. The authors do not discuss it apart from pointing it out.

Given that the proposed algorithm adds a password for the encryption, it seems that the gain in performance is in using a faster, standard encryption algorithm.
If this is the case, then the comparison is not really clear and valid.

Validity of the findings

I am not entirely convinced why the proposed encryption scheme is more secure than existing ones.
The schema is based on password protection first, which is not really an encryption scheme.
The second stage is to use an existing encryption algorithm to encrypt the data.
Adding another password before encryption is hardly making a more secure encryption schema.

Additional comments

In my opinion, the paper lacks a more rigorous explanation of the proposed improvement and a proof why it is better than the existing ones.

Reviewer 2 ·

Basic reporting

Some important references and discussion are missing.

Can we use the proposed approach in YARN?

What is the difference from the following work? why the following work cannot be used in Hadoop
1. Overencryption: Management of Access Control Evolution on Outsourced Data
2.GuardMR: fine-grained security policy enforcement for MapReduce systems
3. Vigiles: Fine-grained access control for mapreduce systems
4. AccountableMR: Toward accountable MapReduce systems
5. TrustMR: Computation integrity assurance system for MapReduce

Why hybrid cloud-based MR system cannot solve the issue you have, if we do not outsource sensitive data?
1. Towards Privacy for MapReduce on Hybrid Clouds Using Information Dispersal Algorithm
2. SEMROD: Secure and Efficient MapReduce Over HybriD Clouds
3. Hybrid storage architecture and efficient MapReduce processing for unstructured data


It will be good to write difference from the following work
1. MtMR: Ensuring MapReduce Computation Integrity with Merkle Tree-based Verifications
2. SEINA: A Stealthy and Effective Internal Attack in Hadoop Systems
3. Security Threats to Hadoop: Data Leakage Attacks and Investigation
4. Privacy-Preserving Secret Shared Computations using MapReduce

Why SGX-based MR system cannot solve the issue that you are trying to deal with?
1. VC3: Trustworthy Data Analytics in the Cloud using SGX
2. M2R: Enabling Stronger Privacy in MapReduce Computation

Experimental design

please include comments given above

Validity of the findings

please include comments given above

Additional comments

please include comments given above

---

## Round 0.3 · accepted · Accept

Both reviewers recommended acceptance, please proofread the final version.

·

Basic reporting

The paper is written clearly and improved very much since the last revisions.

Experimental design

The experiments are satisfactory.

Validity of the findings

no comment

Reviewer 2 ·

Basic reporting

Accept

Experimental design

Accept

Validity of the findings

Accept

Additional comments

Accept